# Does Chytridiomycosis Affect Tree Frog Attachment?

Lisa Nieuwboer [1], Johan L. van Leeuwen [1], An Martel [2], Frank Pasmans [2],
Annemarieke Spitzen-van der Sluijs [3] and Julian K. A. Langowski [1,*]

[1] Experimental Zoology Group, Department of Animal Sciences, Wageningen University & Research, De Elst 1, 6708 WD Wageningen, The Netherlands; lisa.nieuwboer@wur.nl (L.N.); johan.vanleeuwen@wur.nl (J.L.v.L.)
[2] Wildlife Health Ghent, Department of Pathology, Bacteriology and Poultry Diseases, Ghent University, Salisburylaan 133, 9820 Merelbeke, Belgium; An.Martel@UGent.be (A.M.); Frank.Pasmans@UGent.be (F.P.)
[3] Reptile, Amphibian and Fish Conservation Netherlands (RAVON),
P.O. Box 1413, 6501 BK Nijmegen, The Netherlands; a.spitzen@ravon.nl
* Correspondence: julian.langowski@wur.nl

**Abstract:** The pandemic disease chytridiomycosis, caused by the fungus *Batrachochytrium dendrobatidis* (*Bd*), is a major threat to amphibian biodiversity. For most species, the exact mechanisms of chytridiomycosis that lead to negative population dynamics remain uncertain, though mounting evidence suggests that sublethal effects could be an important driver. In this review, we propose that tree frog attachment is a promising case to study the sublethal effects of a *Bd* infection on amphibians. A synthesis of the current knowledge on the functional morphology of the adhesive toe pads of tree frogs, on the underlying mechanisms of tree frog attachment, and on the epidermal pathology of chytridiomycosis substantiates the hypothesis that *Bd*-induced epidermal alterations have the potential to disrupt tree frog attachment. We highlight a series of (biomechanical) experiments to test this hypothesis and to shed some light on the sublethal disease mechanisms of chytridiomycosis. The knowledge generated from such an approach could contribute to future research on *Bd* epidemiology and ultimately to the conservation of the biodiversity of arboreal anurans.

**Keywords:** *Batrachochytrium dendrobatidis*; anura; *Litoria caerulea*; epidermis; bioadhesion; arboreality

## 1. Introduction

The fungal disease chytridiomycosis is a global threat to amphibian biodiversity [1–4]. This pandemic disease has been linked to population declines of hundreds of species, among which many are hylid tree frog species [3]. Curbing this biodiversity loss is crucial, because amphibians are an important link in the food web of many aquatic and terrestrial ecosystems [5]. The design of effective mitigation strategies for chytridiomycosis requires a thorough understanding of the fundamental processes that determine disease impact, many of which are still unknown [6].

Chytridiomycosis is caused by a sustained cutaneous infection with the chytrid fungi *Batrachochytrium dendrobatidis* (henceforth referred to as *Bd*) and *Batrachochytrium salamandrivorans*. Even though infection with both fungi can result in lethal disease and population extirpation, their symptomatology is very different [7]. Because *Bd* is associated with severe anuran population declines, we focus here on *Bd*. *Bd*-infected epidermis thickens due to increased cell division (hyperplasia) and keratin production (hyperkeratosis) [8,9]. Skin thickening hampers regular amphibian skin functioning in vital processes such as transcutaneous respiration and osmoregulation [10], which leads to an electrolyte imbalance and can ultimately result in cardiac arrest [11]. This lethal effect underpins the emblematic *Bd*-caused mass-mortality events that result in rapid population declines or even species extinction [8].

However, for the majority of affected frog species these mass mortality events and the related skin pathology have not been detected [3], and a "more conclusive determination of the role of *Bd* as a causative agent in amphibian declines" [12] (p. 116) is needed. Mounting evidence

suggests that cryptic sublethal effects are important as well in negatively driving population dynamics [13–15]. Several sublethal effects have been identified for adult frogs. For example, the energy required to mount a strong immune response resulted in lower growth rates in infected frogs and toads [14]. Furthermore, *Bd*-infected frogs slough their skin more frequently, which increases the risk of dehydration [16–18]. Sublethal effects may also include altered behavior, with multiple studies reporting higher (re-)capture rates of infected individuals as they were residing in locations where they were easy to spot [13,15,19]. However, the mechanisms that underly the sublethal effects of chytridiomycosis remain largely unknown [20,21], and the relation between *Bd*-induced sublethal effects, reduced individual performance, decreased survival probability, and amphibian fitness is barely understood.

As chytridiomycosis is a cutaneous disease, pathologic alterations of the skin morphology are a likely cause of sublethal effects. This may apply particularly to tree frogs; the intricate morphology of their toe epidermis enables these animals to securely grip various substrates, thus facilitating an arboreal lifestyle. Living above the ground allows these frogs to avoid ground-dwelling predators, to gain access to food sources unreachable to terrestrial anurans, and to use arboreal breeding sites [22].

As *Bd* is known to heavily infect the ventral side of anuran toes [9,23], we hypothesize that *Bd* has the potential to disrupt the functional morphology of tree frog toe pads, thus negatively affecting the attachment performance and fitness of these animals. This proposed novel, biomechanical perspective renders tree frogs an interesting model to study the sublethal effects of *Bd*, and may help to unravel previously overlooked sublethal disease mechanisms.

To substantiate our biomechanical view on studying the sublethal effects of chytridiomycosis, we present here an overview of the current knowledge on the functional morphology of tree frog toe pads and on the skin alterations caused by chytridiomycosis. We further discuss how the functional morphology of tree frog toe pads facilitates the remarkable attachment of these animals, and present hypotheses on how *Bd*-induced skin alterations may affect the fundamental mechanisms of tree frog attachment. Finally, we highlight experiments that will be crucial to unravel the sublethal effects of *Bd* on tree frog attachment, and we elaborate how insights resulting from these experiments could contribute to developing conservation strategies for arboreal anurans.

## 2. Comparing the Morphology of Healthy and *Bd*-Infected Anuran Epidermis

The polyphyletic taxon "tree frogs" can be defined in its widest sense as anurans with an arboreal lifestyle [24]. Arboreality is facilitated by adaptations both in behavior (e.g., limb splaying when hanging on overhanging substrates [25]) and in the morphology of the locomotory apparatus [26–28]. Expanded disc-shaped pads at the digital tips are the most prominent morphological adaptation of tree frogs to arboreality (Figure 1A). Throughout this manuscript, we refer to these structures as "toe pads". Several superficial and internal morphological characteristics of the toe pads synergistically enable the remarkable attachment of tree frogs [29,30]. As basis for the later discussion of the potential effects of *Bd* on the attachment of tree frogs, we provide in the following section an account of the functional morphology of tree frog toe pads, and of *Bd*-induced skin alterations.

### 2.1. Functional Morphology of Tree Frog Toe Pads

The tip of a tree frog's digit consists of the terminal phalanx (i.e., the most distal digital bone), a subcutaneous lymph space, and dermal and epidermal tissues (Figure 1B; [31,32]). The terminal phalanx connects via an intercalary element, ligaments, and tendons to more basal parts of the musculoskeletal system. The terminal phalanx is separated from the adhesive ventral skin by a lymph-filled space ("lymph space") and a subdermal cluster of mucus glands that open to the ventral pad surface [33]. Lymph space, mucus gland cluster, and ventral skin together constitute the actual toe pad.

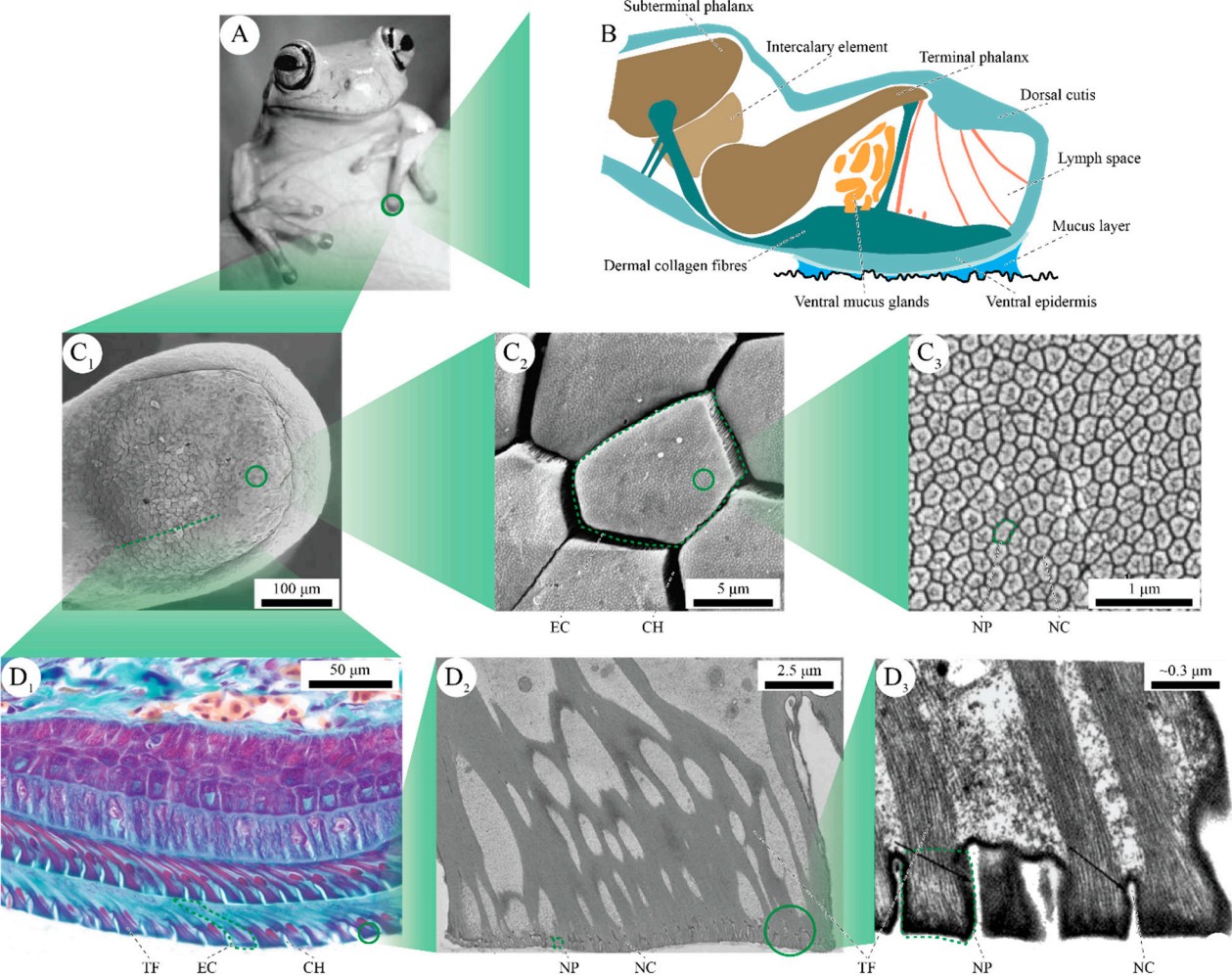

**Figure 1.** Functional morphology of the adhesive toe pads of healthy hylid tree frogs. (**A**) *Litoria caerulea* with toes in contact with a substrate. (**B**) Schematic depiction of the functionally relevant structures in the tip of a tree frog digit, lateral view. (**C**$_{1-3}$) Superficial morphology of the ventral pad epidermis of *L. caerulea*, scanning electron micrographs in ventral view. (**C**$_1$) Toe pad. (**C**$_2$) Polygonal epidermal cell surrounded by channels. (**C**$_3$) Nanopillars. (**D**$_{1-3}$) Internal morphology of the ventral pad epidermis shown for *Hyla cinerea* (**D**$_{1,3}$) and *L. caerulea* (**D**$_2$), lateral view, left—proximal, right—distal. (**D**$_1$) Section through the ventral epidermis stained with Crossmon's light green trichrome including Mayer's haematoxylin and Alcian blue. (**D**$_2$) Apical freestanding part of an epidermal cell. (**D**$_3$) Nanopillars with tonofilaments. CH, channel between epidermal cells; EC, epidermal cell; NC, channel between nanopillars; NP, nanopillars; TF, tonofilaments. **A** and **C**$_{1-3}$ modified after [34]; **B** and **D**$_1$ modified after [33]; **D**$_2$ modified after [35]; **D**$_3$ modified after [31]. All figures were reproduced with permission.

The ventral pad skin consists of dermal collagen fibers that link the epidermis to the subterminal phalanx [31,33], and of the adhesive ventral epidermis that contacts the underlying substrate. The ventral pad epidermis is columnar and relatively thick (ca. 100–220 µm) if compared to squamous, non-adhesive epidermis [33,36,37]. The number of epidermal cell layers varies interspecifically, with three cell layers reported for *Litoria caerulea* [38] and six layers for *Hyla cinerea* [31]. The columnar cell arrangement becomes clearer towards the apical epidermal cell layer (Figure 1D$_1$; [31]).

The micro- and nanoscopic topography of the superficial cell layer of the ventral epidermis is the most prominent morphological feature of tree frog toe pads (Figure 1C$_{1-3}$). Whereas the basal parts of these cells are tightly interconnected, the apical parts are freestanding and separated from each other by an intercellular channel network. The cells are skewed such that the freestanding part is located more distally than the interconnected part. The apical surfaces of the freestanding pillar-like cell parts have polygonal outlines

and are typically around 10 μm high and wide [29]. The resulting pillar-channel pattern is hierarchical: nanoscopic protrusions on the apical surface of each cell form an array of so-called "nanopillars", which also are separated by channels. Single nanopillars are ca. 300 nm high and wide [29].

In addition, the internal morphology of the ventral toe pad epidermis differs from that of regular amphibian epidermis (Figure $1D_{1-3}$). The apical epidermal layers have been repeatedly described as "keratinized" [39]. This keratinization is to equate with tonofilaments that traverse the epidermal cells along the apical-basal cell axis [31,35]. In the superficial cell layer (i.e., stratum corneum), the tonofilaments form a distinct fiber network: they are bundled in the basal–proximal corner of the interconnected cell part, diverge from there towards the apical cell surface, and connect approximately perpendicularly to the apical surfaces of the single nanopillars [31,35].

### 2.2. Epidermal Morphology of Bd-Infected Frogs

The infection cycle of *Bd* is well understood and appears to be similar across various anuran families (Hylidae, Dendrobatidae, Ranidae, and Lymnodinastidae [7,40–42]). Current knowledge on morphological changes of anuran skin upon *Bd* infection originates primarily from studies on the genus *Litoria*. Previous studies largely focused on the ventral abdomen and interdigital skin [9,40–43], and exclusively relied on 2D techniques such as histology and electron microscopy to study the morphology of *Bd*-infected epidermis (Table 1). Most studies were performed on severely infected animals that died from chytridiomycosis.

Generally, the zoospores of *Bd* attach to the epidermal cells of an anuran host and then form germination tubes to infect more basal epidermal cell layers [40]. Once inside the host cell, a zoospore develops into a small thallus with a new spore-forming body (zoosporangium). Meanwhile, the cells migrate outwards with the regular molting cycle to form the new apical epidermal cell layer, thus bringing the sporangium to the surface. Finally, the zoosporangium releases new spores through a discharge tube that protrudes outwards through the cell membrane [7,41]. The presence of up to three sporangia per cell [35] can induce severe morphological alterations in the infected cell as well as in neighboring epidermal cells.

**Table 1.** Overview of *Bd*-induced epidermal alterations in hylid tree frogs (top) and other anuran species (bottom).

| Ref. | Species | Skin Location | Method | Infection Severity | Observed Skin Alterations |
|---|---|---|---|---|---|
| [43] | *Litoria caerulea* | Abdominal pelvic patch | Histology | Varying (approximated by zoospore load) | - Cutaneous erythema, histological lesions and thinning of the skin |
| [8] | *L. caerulea* | Digital | Histology | High* | - Stratum corneum thickened from 2–5 μm to 60 μm <br> - Mature sporangia diameter = 12–20 μm |
| [9] | *L. caerulea* | Ventral skin | Histology | High* | - Hyperplasia thickens epidermis from 25–60 μm (4–7 layers) to up to 125 μm (13 layers) |
| [44] | *L. caerulea* | Ventral abdomen, pelvis, and hindlimbs | Histology | High* | - Cutaneous lesions characterized by hyperplasia and hyperkeratosis |
| [41] | *L. gracilenta* | Toe clip | TEM | Varying (one animal with mild infection, one died from infection) | - Up to 4 extra keratinized skin layers in stratum corneum (mild infection) <br> - "Fibrillar zone" (2.5 μm wide) around sporangia |

**Table 1.** *Cont.*

| Ref. | Species | Skin Location | Method | Infection Severity | Observed Skin Alterations |
|---|---|---|---|---|---|
| [41] | *L. lesueuri* | Interdigital | SEM | Unknown | - Bulging epidermal cells with discharge tubes covering ca. 10% of the cell surface<br>- Up to 3 sporangia in an epidermal cell<br>- Sporangia diameter up to 40 μm |
| [45] | *Pseudacris regilla* | Ventral pelvic patch, foot webbing | Histology | Varying (five animals with non-lethal infection, one died from infection) | - Hyperkeratosis and hyperplasia across the whole skin (lethally affected individual) or in discrete patches (non-lethally affected individuals)<br>- Cutaneous lesions with high density of *Bd*-sporangia |
| [40] | *Lithobates catesbeianus* | Interdigital | EM | Low (study of the infection process) | - Clear keratin-free zones of host cytoplasm around zoosporangium<br>- Premature keratinization of 2–3 outer cell layers |
| [46] | *Dendrobates spec.* | Various | Histology | High * | - Hyperkeratosis, hyperplasia, and/or hypertrophy of nonkeratinized epidermal cells |
| [42] | *Dendrobates azureus* | Various | Histology | High * | - Germination tubes extend towards the skin surface |

TEM: transmission electron microscopy, SEM: scanning electron microscopy, EM: electron microscopy. *: Animals died from *Bd* infection.

Superficially, infected epidermal cells have been observed to bulge outwards, with discharge tubes covering up to 10% of the cell surface in *Litoria lesueuri* (Figure 2B$_2$; [41]). Internally, the tonofilaments and other epidermal cell organelles are rearranged and leave a ca. 2.5 μm wide zone of host cytoplasm around the zoosporangium (Figure 2B$_3$; [40,41]).

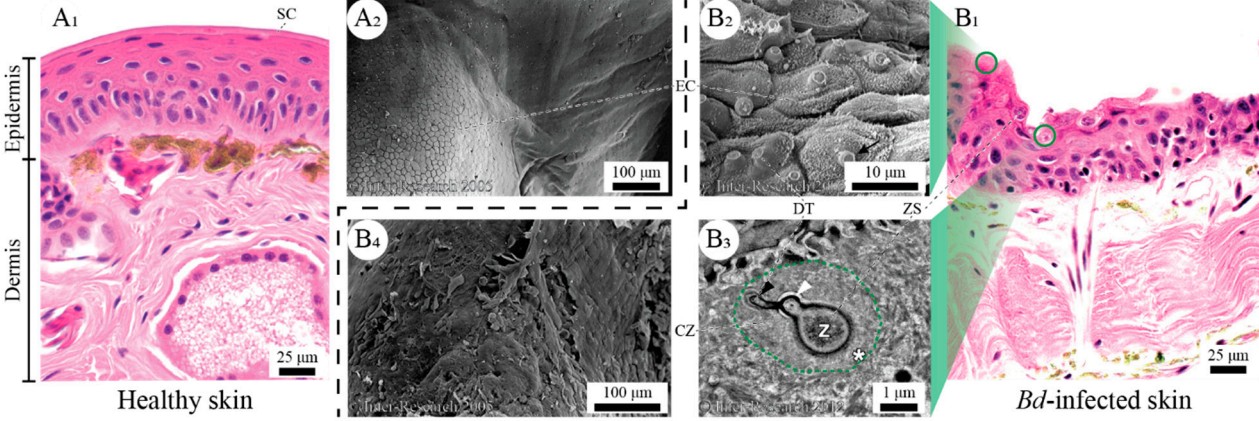

**Figure 2.** Differences in the morphology of (**A**) healthy and (**B**) *Bd*-infected anuran skin. (**A$_1$**) Healthy epidermis is multilayered and covered by an intact keratinized stratum corneum; Hematoxylin-eosin-stained skin section of *Litoria caerulea*. (**A$_2$**) The ventral epidermis of tree frog toe pads shows a characteristic pillar-channel micropattern (see also Figure 1); scanning electron micrograph (SEM) of the ventral toe pad surface of *L. caerulea*. (**B$_1$**) Infected epidermis contains *Bd* zoosporangia, can be thinned (shown here) or thickened, and suffer from desquamatization (i.e., loss of the stratum corneum); *L. caerulea*. (**B$_2$**) Discharge tubes of the zoosporangia protrude from the superficial epidermal cells, which also show convex bulging of the apical cell surfaces; SEM of the toe skin of *Litoria lesueuri*. (**B$_3$**) The architecture of epidermal tonofilaments is rearranged around the zoosporangia, leaving a "clear zone" around the zoosporangia; transmission electron micrograph of the interdigital skin of *Lithobates catesbeianus*. (**B$_4$**) *Bd*-infected foot skin is devoid of micropatterning; SEM of the foot skin of *L. lesueuri*. CZ: clear zone surrounded by rearranged tonofilaments; DT: discharge tube; SC: stratum corneum; ZS: zoospore. **A$_1$** and **B$_1$** modified after [47,48]; **A$_2$** and **B$_{2,4}$** modified after [41]; **B$_3$** modified after [40]. All figures were reproduced with permission.

The presence of the zoosporangia in a host cell induces cell division (hyperplasia) in neighboring epidermal cells, resulting in local thickening of the epidermis [8,9,42,45,46]. For example, in *L. caerulea* the stratum corneum (i.e., the apical epidermal cell layer) can thicken from 2–5 μm to 60 μm [6]. *Bd*-infected epidermis is not only thicker, but also more heavily keratinized than healthy skin (hyperkeratosis). Up to four additional keratinized cell layers were observed in the tree frog *Litoria gracilenta* with only a mild infection [41]. Next to skin thickening, skin may also become locally thinner because of lesions and/or increased shedding (Figure 2B$_1$; [42,43,45]).

## 3. Potential Sublethal Effects of Chytridiomycosis on Tree Frog Attachment

### *3.1. Mechanisms of Tree Frog Attachment*

The attachment of tree frogs relies on a combination of physicochemical mechanisms that are embodied in the morphology of the locomotory apparatus of these animals. Most importantly, the toe pads of tree frogs presumably adhere to a substrate by the generation of intermolecular van der Waals forces and of mucus-mediated capillary adhesion. For a comprehensive account of the fundamental mechanisms of tree frog attachment, we refer to [29,30,49]. Below, we focus on mechanisms that are facilitated specifically by the functional morphology of the toe pad epidermis, and hence may be disturbed by chytridiomycosis.

Strong grip relies on a large area of intimate contact between adhesive and substrate. Natural substrates are inherently rough, which can hinder the formation of such close contact [50]. Nonetheless, several adaptations of the adhesive epidermis of frog pads facilitate closure of the gap between toe and substrate down to a gap width of a few nanometers [34]. The toe pads consist of soft materials (e.g., the lymph space) that can be deformed easily and thus conform to macroscopic substrate irregularities (Figure 3A$_1$; [29,39]). Moreover, the hierarchical patterning of the ventral pad surface reduces the effective stiffness of the skin, and single micro- and nanopillars presumably can conform to substrate irregularities on the corresponding size levels (Figure 3A$_2$; [51]).

Next to substrate roughness, the mucus layer covering amphibian skin should also impede strong grip in tree frogs. The mucus potentially acts as a lubricant that widens the pad–substrate gap and thus reduces the area of close contact [29]. However, the interconnected channels between the epidermal cells and nanopillars thereon likely serve as a drainage system that allows removal of mucus from the interstitial space, closure of the pad–substrate gap, and generation of strong attachment (Figure 3A$_3$; [29,52,53]).

Once a large area of close contact has been established, this area needs to be maintained to avoid detachment of the toe pads when loaded. Detachment occurs when mechanical loading of the pad–substrate interface results in the buildup of contact stresses that exceed the maximum possible adhesive stress [54]. Such detachment of frog pads can be approximately described by peeling theory [25,55], analogously to the gradual peeling-off of a strip of sticky tape. However, the adhered epidermis of a frog pad is not loaded one-sidedly, as is sticky tape, but through the fibrous dermal collagen layer and epidermal tonofilaments [33]. This internal fiber reinforcement potentially promotes a uniform distribution of contact stresses and thus counteracts detachment (Figure 3A$_4$; [30]).

Besides a favorable contact stress distribution, the tonofilaments also strengthen the epidermis mechanically. During rapid locomotory maneuvers such as jumping, tree frog toe pads are exposed to substantial loads of up to a multiple of the animals' body weight [56]. An intact tonofilament morphology is crucial to maintain structural integrity of the toe pad epidermis during such peak loading events.

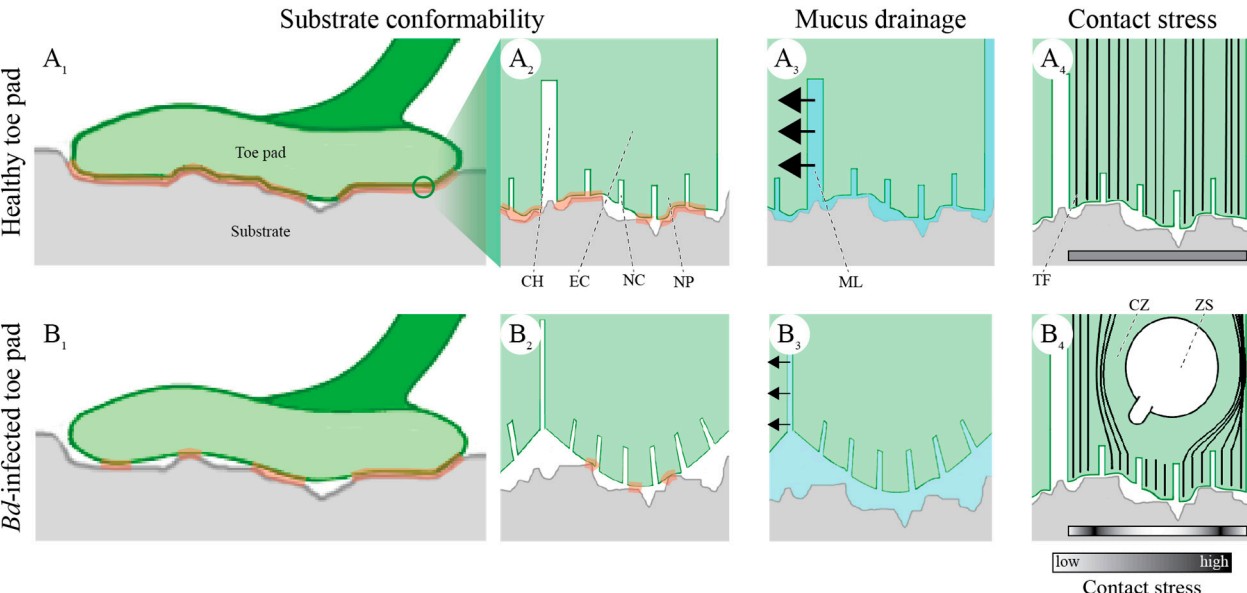

**Figure 3.** Schematic depiction of (**A**) some of the attachment mechanisms embodied in the healthy epidermis of tree frog toe pads, and (**B**) the hypothesized disruptions of these mechanisms upon *Bd* infection. (**A**$_{1,2}$) The softness of healthy toe pads and the hierarchical pillar-channel pattern on the pad surface facilitate the formation of a large area of close contact (red) on macrorough and micro- to nanorough substrates, respectively. (**A**$_3$) The interconnected network of channels in between epidermal cells and nanopillars enables effective drainage of mucus and other interstitial liquids (large arrows), thus supporting the formation of close contact. (**A**$_4$) An intact tonofilament network possibly ensures an approximately uniform distribution of contact stresses across the contact surface. (**B**$_1$) Hyperkeratosis and hyperplasia may stiffen the toe pad and reduce conformability to macrorough substrates. (**B**$_2$) Convex bulging of epidermal cells could hamper close contact formation to (sub-)microscopically rough substrates. (**B**$_3$) Narrowed channels result in less effective mucus drainage (small arrows) and disable close contact formation. (**B**$_4$) Rearrangement of tonofilaments around zoosporangia disrupts a uniform contact stress distribution and may increase local stress maxima. CH, channel between epidermal cells; CZ, clear zone surrounded by rearranged tonofilaments; EC, epidermal cell; ML, mucus layer; NC, channel between nanopillars; NP, nanopillars; TF, tonofilaments; ZS, zoospore.

### 3.2. Bd-Induced Disruptions of Tree Frog Attachment

*Bd*-induced epidermal alterations could affect the functional morphology of tree frog toe pads and, consequently, the mechanisms of tree frog attachment in several ways. Below, we present possible effects of changes in the superficial and internal morphology of anuran toe pad epidermis on the aforementioned mechanisms of tree frog attachment. These effects may occur separately or in (possibly synergetic) combination.

*Bd*-induced epidermal alterations could reduce the conformability of tree frog toe pads to rough substrates, thus hindering the formation of a large area of close contact and deteriorating the attachment performance of these animals. An increased keratin content (i.e., hyperkeratosis) and the pathological thickening (i.e., hyperplasia) of *Bd*-infected epidermis could stiffen the normally soft and flexible pad, which would inhibit conformation of the pad to macroscopic topological features of a substrate (Figure 3B$_1$). On a microscopic level, the apical bulging of *Bd*-infected epidermal cells may reduce the conformability of these pillar-like structures to microscopic substrate irregularities (Figure 3B$_2$). Further reduction in conformability of the pad surface may arise from the fungal discharge tubes that protrude to the apical surfaces of infected cells. The pad surface area covered with these tubes may contribute only partially, or not at all, to the effective contact area.

Close contact formation also depends on effective drainage of the mucus layer from the gap between toe pad and substrate. Pathological deformation (e.g., bulging) of the epidermal cells may narrow the intercellular channels and thus prevent effective mucus drainage. A wider pad–substrate gap due to poor mucus drainage may result in a reduced

area of close contact as well as weakened van der Waals interactions between toe pad and substrate (Figure 3A$_3$).

The network of tonofilaments inside the epidermal cells, which provides mechanical strength to frog pads and possibly homogenizes the distribution of contact stresses, could also be affected by *Bd*. The redistribution of the tonofilaments and formation of a clear zone around zoosporangia is likely to affect the transmission of mechanical loads between the adhesive epidermis and the rest of the animal's body, and may lead to an altered distribution of contact stresses over the apical surfaces of individual cells. As a result, locally concentrated and increased contact stresses may cause local peeling of single cell–substrate contacts and thus reduce the effective attachment performance of frog pads (Figure 3B$_2$).

Finally, a pathological tonofilament morphology could reduce the mechanical strength of the toe pads, thus weakening the pad's structural integrity and disabling its ability to withstand mechanical peak loads during dynamic events such as jumping [56].

## 4. How to Test If *Bd* Affects Tree Frog Attachment?

Albeit based on current knowledge on the adhesive system of tree frogs and on the epidermal pathology of *Bd*, an interdisciplinary set of approaches ranging from in vitro observations to in vivo studies is required to verify the hypothesized effects of *Bd* on tree frog attachment. Most importantly, a detailed analysis of the changes in functional toe pad morphology upon *Bd*-infection is needed. As we show in Table 1, previous studies provide only a partial picture of the morphology of *Bd*-infected anuran epidermis in tree frogs and other anurans. In future studies, a thorough description of the anatomical origin of analyzed skin samples would help to complete our knowledge on morphological symptoms of chytridiomycosis. Surprisingly, we are not aware of any systematic comparison of the morphology of healthy and *Bd*-infected epidermis. Such a comparative approach is crucial to fully understand *Bd*-induced changes of anuran epidermis on the toe pads as well as other body parts.

While Berger et al. [8] showed that chytridiomycosis does affect the foot epidermis of hylid tree frogs, a detailed analysis of pathological alterations of the ventral toe pad skin is still unavailable. We propose a three-dimensional, quantitative, and hierarchical comparative analysis of healthy and *Bd*-infected toe pads from the level of single epidermal cells to whole toe pad tissue, with focus on both the superficial and internal toe pad morphology. Atomic force microscopy (AFM) may be used to unravel the so-far unstudied pathological alterations of the nanopillar morphology. The combination of AFM, classical histology, scanning electron microscopy (SEM), and synchrotron micro-computer-tomography (μ-CT) could significantly expand our understanding of the morphological effects of chytridiomycosis beyond mere measurements of skin thickness (see Table 1). Moreover, most previous studies were done on severely infected animals that deceased from chytridiomycosis, infection loads were rarely reported, and the dynamics of *Bd*-related pathological skin alterations, consequentially, are barely described. Hence, the aforementioned measurements should be done in a longitudinal manner throughout the progression of chytridiomycosis.

Mechano-indentation techniques, which are commonly used in material sciences, could be used to assess whether *Bd*-induced epidermal alterations translate into changes in properties that are relevant to attachment. For example, microindentation [39] and nanoindentation using AFM [57,58] could be combined to compare the stiffness of healthy and *Bd*-infected toe pads on different size levels. An increased stiffness of *Bd*-infected pads would strongly support the hypothesized mechanism of impaired substrate conformability. Nano-indentation could also be used to test if the presence of *Bd* discharge tubes affects the epidermal surface not only structurally, but also mechanically.

Biomechanical experiments are necessary to assess if and how *Bd*-induced alterations in the functional pad morphology lead to changes in the attachment performance of individual animals. For example, a rotation platform setup or force-transducer—both are frequently used techniques in bioadhesion research [29]—allow measurement of the attachment forces generated by healthy and infected animals. Ideally, such experiments

provide parameters that can also be measured or observed in the wild, such as behavioral or microhabitat changes. Complementary field studies, such as comparing the finding location of healthy and *Bd*-infected tree frog species, could further substantiate how a reduced attachment performance leads to reduced fitness and population declines. We expect that *Bd*-infected individuals may be less frequently found in the higher levels of their arboreal habitats. *L. caerulea* could be a suitable study species for such an approach, because it is—next to its status as model species for bioadhesion and *Bd* research (see below)—widely distributed, abundant, and moderately affected by *Bd* [59], which facilitates long-term observations for large sample numbers on the occurrence of sublethal effects.

Next to the aforementioned changes of the toe pad morphology, a *Bd* infection also alters the composition of amphibian mucus, which plays an important role in the defense against pathogens [60] and possibly also in tree frog attachment. For example, in *Xenopus laevis*, an initial *Bd* infection leads to an increased production of antifungal skin peptides, and after first infection the mucus contains various antibodies that protect the frogs from a new infection [61,62]. Although the role of the mucus in the attachment of tree frogs is under debate [29] and the properties of healthy tree frog mucus are only partially known [63], a future comprehensive analysis of healthy and *Bd*-infected toe pad mucus would be useful.

*L. caerulea* appears to be a promising model species for studying the sublethal effects of chytridiomycosis on the attachment of tree frogs. As a docile, common, and robust species, *L. caerulea* is well suited for laboratory experiments, and the toe pad morphology and underlying attachment mechanisms are well characterized [29,34,64]. Furthermore, *L. caerulea* shows typical chytridiomycosis upon infection [12], and—as shown in Table 1— is a model species for *Bd* epidemiology.

## 5. Conclusions and Perspectives

The amphibian epidermis is an important physiological and (bio-)mechanical organ that acts as a physicochemical interface with the environment. Therefore, disease-driven alterations of anuran skin may affect the physiology as well as biomechanics of these animals. Whereas the often-lethal effects of a *Bd* infection on the physiology of anuran hosts have been largely unraveled in the past two decades [7,12,65], potential sublethal and more cryptic effects on the adhesive performance and consequentially fitness of these animals are virtually unstudied.

In this review, we show that tree frogs are a promising case to study the sublethal effects of chytridiomycosis on anuran hosts, because the epidermis of tree frogs is crucial for the attachment and arboreal life of these animals. Moreover, tree frogs represent a large part of amphibian biodiversity. The Hylidae alone account for one eighth of all amphibian species [66], and members of various other anuran (and even batrachian) families also possess a hierarchically patterned ventral toe pad epidermis, which cofacilitates an arboreal lifestyle [64]. The extensive knowledge on the functional morphology of the adhesive toe pads of tree frogs (Section 2.1), the presence of multiple *Bd*-induced epidermal alterations with the potential to disrupt the pad functionality in attachment (Sections 2.2 and 3.2), and the availability of various experimental methods for future studies (Section 4) further corroborate the relevance of tree frogs as a model to study the sublethal effects of chytridiomycosis.

Studying the sublethal effects of a *Bd* infection on tree frog attachment from a biomechanical point of view could provide valuable insights into how chytridiomycosis causes a reduced fitness of individual animals and, consequently, whole population declines. Obtaining such an understanding could be an important step towards the conservation of tree frog, as well as other amphibian, species. For example, validation of the hypotheses presented here may stimulate the inclusion of ecology- and natural-history-related factors in future models of *Bd* susceptibility, as compared to current models that focus on climate-related factors [12,67,68]. The more accurate prediction of amphibian species endangered by *Bd*, taking into account also sublethal long-term disease mechanisms, can help to identify

particularly threatened frog species, which may be entered into captive breeding programs. Interestingly, the inclusion of life history traits in models of *Bd* susceptibility indicates that stream-dwelling tree frogs are more vulnerable than exclusively arboreal species [69,70]. While this may partially be explained by the waterborne dispersal of *Bd*, a sublethal effect on the attachment of stream-dwelling frogs may offer an additional explanation on their increased vulnerability. In stream-dwelling frogs, a ventral epidermal toe pad morphology similar to that of tree frogs enables strong attachment to rocks and other substrates in fast-flowing water [71]. *Bd*-related alteration of that morphology may impair attachment and thus partially explain the *Bd* susceptibility of stream-dwelling frogs. Implementation of such potential interaction effects between life history traits of *Bd* and hosts could help to optimize models of *Bd* susceptibility.

The conservation of frogs is crucial not only from an ecological perspective, but also because of the relevance of this group of animals in biotechnology and biomimetics. Amphibian mucus is an important source of biomolecules with potential for the development of novel therapeutics [72,73] and functional materials [74]. Tree frog toe pads are a prominent model for the development of biomimetic adhesive surfaces [30] and robotic gripping systems [75]. Interestingly, studying the sublethal effects of chytridiomycosis on tree frog attachment may not only ultimately contribute to the conservation of this remarkable group of animals, but *Bd*-infected pads also represent a unique disease model to test hypotheses on the functional relevance of several pad structures and on the fundamentals of tree frog attachment.

**Author Contributions:** Conceptualization, J.K.A.L. and L.N.; literature review, L.N. and J.K.A.L.; writing—original draft preparation, J.K.A.L. and L.N.; writing—review and editing, J.K.A.L., L.N., J.L.v.L., F.P., A.M., A.S.-v.d.S.; visualization, J.K.A.L. and L.N. All authors have read and agreed to the published version of the manuscript.

**Funding:** JKAL was funded within the Soft Robotics Consortium with project number 4TU-UIT-335 (internal: 4162504200), which is financed by the 4TU.Federation.

**Conflicts of Interest:** The authors declare no conflict of interest.

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
