# Peer review of "Does Chytridiomycosis Affect Tree Frog Attachment?"

_diversity, doi:10.3390/d13060262_

Round 1

Reviewer 1 Report

The authors presented a comprehensive review paper that provides an experimental framework to test potential links (or causal effects) between the chytrid fungus Batrachochytrium dendrobatidis (Bd) and the decline of some amphibian species through sublethal effects affecting frog attachment. The authors first provided a detailed and summarized overview of amphibian chytridiomycosis and a thorough description of the anatomy and physiology of toe pads of treefrogs. Finally, the authors addressed potential mechanisms to test experimental hypotheses. I enjoyed reading the paper. Overall, the paper is very well-written, and the review is highly detailed. I consider this MS presents very valuable information that can be used for a better understanding of the enigmatic global declines that have affected some amphibian species, especially those with particular life-history traits. I have a few minor concerns that might be addressed by the authors and provided a few citations that I think should be added.

Introduction: I consider the paper can start with a broader focus. For some readers that could be unaware of amphibian chytridiomycosis, it would be useful to establish in the first paragraph that this disease can also occur through infection with B. salamandrivorans (Bsal). Then the authors may indicate that both types of amphibian chytridiomycosis have different symptomatology and that they will focus on Bd because it has been associated with large declines in anuran species. The second sentence in the first paragraph should be slightly amended. Bd has been ‘linked’ with the decline of at least 501 species; it’s not clear that all species declined because of Bd as stated by the authors. Particularly, there is a paper (Lambert et al., 2020) that disagrees with the way those 501 species were accounted for. I suggest rephrasing a little the paragraph and citing also the alternative paper to provide a more neutral scope to this particular subject. Finally, the authors might suggest that some studies have identified life-history traits that might be associated with Bd-driven declines (e.g., Hero et al., 2005; Zumbado-Ulate et al., 2019) and then focus on tree frogs.

Table 1. I recommend adding to this table the study of Puschendorf & Bolaños (2006) on histological detection of Bd on the species Craugastor fitzingeri (former Eleutherodactylus fitzingeri) since this species is a direct-developing, semiarboreal species (but also common in streams) in the tropics of Central America and Colombia. This species belongs to a family (Craugastoridae) that has experienced catastrophic declines in the region, especially in stream-dwelling species.

Conclusions: It would be recommendable to discuss in some degree what might be happening with tree frogs that have been identified as potential Bd-reservoir species, for example, Pseudacris regilla in the US (Reeder et al., 2012). Similarly, some genera of treefrogs in Mesoamerica seem to be not affected at all by Bd (e.g., Smilisca, Dendropsophus) and I am sure the authors know more examples of treefrog species from other latitudes. What anatomic or behavioral adaptations of these species are minimizing the negative effects of Bd on frog attachment? Finally, stream-dwelling treefrogs seem to be more vulnerable to Bd than arboreal treefrogs, so I think the authors could establish some potential mechanism to identify if tree frog attachment (and how) is also affecting the most to treefrogs that spend most of their life moving in non-arboreal substrates (e.g., rocks, sand, etc.).

Hero, J.-M., Williams, S. E., & Magnusson, W. E. (2005). Ecological traits of declining amphibians in upland areas of eastern Australia. Journal of Zoology, 267(3), 221–232. https://doi.org/10.1017/S0952836905007296

Lambert, M. R., Womack, M. C., Byrne, A. Q., Hernández-Gómez, O., Noss, C. F., Rothstein, A. P., Blackburn, D. C., Collins, J. P., Crump, M. L., Koo, M. S., Nanjappa, P., Rollins-Smith, L., Vredenburg, V. T., & Rosenblum, E. B. (2020). Comment on “Amphibian fungal panzootic causes catastrophic and ongoing loss of biodiversity.” Science, 367(6484), eaay1838. https://doi.org/10.1126/science.aay1838

Puschendorf, R., & Bolaños, F. (2006). Detection of Batrachochytrium dendrobatidis in Eleutherodactylus fitzingeri: Effects of skin sample location and histologic stain. Journal of Wildlife Diseases, 42(2), 301–306.

Reeder, N. M. M., Pessier, A. P., & Vredenburg, V. T. (2012). A reservoir species for the emerging amphibian pathogen Batrachochytrium dendrobatidis thrives in a landscape decimated by disease. PLOS ONE, 7(3), e33567. https://doi.org/10.1371/journal.pone.0033567

Zumbado-Ulate, H., Nelson, K. N., García-Rodríguez, A., Chaves, G., Arias, E., Bolaños, F., Whitfield, S. M., & Searle, C. L. (2019). Endemic infection of Batrachochytrium dendrobatidis in Costa Rica: Implications for amphibian conservation at regional and species level. Diversity, 11(8), 129. https://doi.org/10.3390/d11080129

Author Response

Dear reviewer,

Thank you very much for your review of our manuscript “Does chytridiomycosis affect tree frog attachment?”. We also thank you for your careful attention to the manuscript and your valuable comments that helped us to improve the final version. Below we provide a point-by-point response to your comments in sequential order.

Yours sincerely,

Lisa Nieuwboer, Julian Langowski

  1. I consider the paper can start with a broader focus. For some readers that could be unaware of amphibian chytridiomycosis, it would be useful to establish in the first paragraph that this disease can also occur through infection with B. salamandrivorans (Bsal). Then the authors may indicate that both types of amphibian chytridiomycosis have different symptomatology and that they will focus on Bd because it has been associated with large declines in anuran species.

We widened the focus of the manuscript, and included in the introduction a short section on the two types of fungi.

  1. The second sentence in the first paragraph should be slightly amended. Bd has been ‘linked’ with the decline of at least 501 species; it’s not clear that all species declined because of Bd as stated by the authors. Particularly, there is a paper (Lambert et al., 2020) that disagrees with the way those 501 species were accounted for. I suggest rephrasing a little the paragraph and citing also the alternative paper to provide a more neutral scope to this particular subject.

We agree with the reviewer and rephrased the (estimated) number of affected species more carefully, as different authors and papers provide very different estimates of affected species. We modified the sentence to include the suggested word “linked” and also cited the alternative paper (Lambert et al. 2020).

  1. Finally, the authors might suggest that some studies have identified life-history traits that might be associated with Bd-driven declines (e.g., Hero et al., 2005; Zumbado-Ulate et al., 2019) and then focus on tree frogs.

  We agree that the connection between life-history traits and Bd-susceptibility is very interesting and relates to this study, as arboreality is an example of a life-history trait. However, we feel that including this topic in the introduction distracts from the main message that epidermal alterations may lead to a sublethal effect on tree frog attachment. Therefore, we chose to elaborate on life-history-traits and how they relate to sublethal effects on tree frog attachment in the discussion (see also comment 6).  

  1. I recommend adding to this table the study of Puschendorf & Bolaños (2006) on histological detection of Bd on the species Craugastor fitzingeri (former Eleutherodactylus fitzingeri) since this species is a direct-developing, semiarboreal species (but also common in streams) in the tropics of Central America and Colombia.

We thank the reviewer for this suggestion. However we did not include this study in table 1, because that table contains only studies on Bd-induced epidermal alterations, in which images of the pathological epidermis (through histology or other methods such as SEM/TEM) are shown. Puschendorf and Bolaños (2006) unfortunately do not include any histological images in their paper, and therefore provide no detailed knowledge on the morphology of infected epidermis. Nevertheless, the results from Puschendorf and Bolaños (2006) emphasise that Bd heavily infects anuran toes and therefore we included this study as a reference in the introduction to support this statement. Furthermore, the body of literature suggested by the reviewer does include a paper with histological images of Bd-infected epidermis (Reeder, Pessier, and Vredenburg, 2012), and we included this study in table 1.

  1. It would be recommendable to discuss in some degree what might be happening with tree frogs that have been identified as potential Bd-reservoir species, for example, Pseudacris regilla in the US (Reeder et al., 2012). Similarly, some genera of treefrogs in Mesoamerica seem to be not affected at all by Bd (e.g., Smilisca, Dendropsophus) and I am sure the authors know more examples of treefrog species from other latitudes. What anatomic or behavioral adaptations of these species are minimizing the negative effects of Bd on frog attachment?

We agree that the identification of some tree frog species as reservoir species is relevant for the hypotheses developed in this review. However, the behavioural and/or life history characteristics of reservoir species are not fully understood (Smith et al., 2007, Diseases of aquatic organisms, Searle et al., 2011, Conservation Biology), and not limited to tree frogs (Woodhams et al., 2008, Herpetological Review). In this review, we aim to substantiate the hypothesis that some tree frog species experience non-lethal effects in their attachment due to Bd. Hypothesizing on how in a 2nd step reservoir species avoid/minimise these sublethal effects appears to be beyond the current scientific knowledge and thus too speculative. Therefore, we refrain from elaborating on this topic.

  1. Finally, stream-dwelling treefrogs seem to be more vulnerable to Bd than arboreal treefrogs, so I think the authors could establish some potential mechanism to identify if tree frog attachment (and how) is also affecting the most to treefrogs that spend most of their life moving in non-arboreal substrates (e.g., rocks, sand, etc.).

We added a section on life-history-traits related to high Bd-susceptibility—particularly on stream-dwelling tree frogs—to the “conclusion and perspectives” part of the manuscript. In this section we discuss how stream-dwelling frogs could be affected by a sublethal effect on attachment and we thank the reviewer for bringing this to our attention.

Reviewer 2 Report

This is a thoroughly interesting review that predicts that chytridiomycosis significantly affects tree frog adhesion and will thus be an important sublethal component of this disease in tree frogs.  My only disappointment is that the section on Bd-induced disruptions of tree frog attachment is theoretical and not based on actual data.  I have no reason to disagree with the conclusions of this section, but conclusions based on actual data would have significantly strengthened the overall conclusions.

Minor comments:

Abstract last sentence:  I think the final conclusions are rather over-optimistic.  Certainly, the findings will increase our understanding of sub-lethal components of chytridiomycosis, but the design of mitigating strategies leading to the conservation of tree frog biodiversity seem to me, sadly, overoptimistic.

  1. Introduction, paragraph 5: Good point; a biomechanical study of the effects of chytridiomycosis on toe pad morphology and function should, indeed, produce useful conclusions on the sub-lethal effects of this disease in tree frogs.

2.2, 2nd paragraph: Isn’t the word ‘thallus’ restricted to plants?

3.1, 1st paragraph: You overemphasise the role of van der Waals forces (see Kappl et al., 2016, Bioinspiration & Bioimimetics) and omit the role of viscosity-dependent hydrodynamic forces (Federle et al., 2006).  The reference to mucus-mediated capillary adhesion is of course appropriate!

3.1, 4th paragraph: You discuss mechanisms that counteract detachment, yet such mechanisms need to be overcome each time the frog jumps or takes a step.

3.2. All very interesting, but I wish you had some actual data!

4, last paragraph: Add ref. 31 alongside refs 26 & 59 as it is a key paper.

References:  The paper is well referenced.

Author Response

Dear reviewer,

Thank you very much for your review of our manuscript “Does chytridiomycosis affect tree frog attachment?”. We also thank you for your careful attention to the manuscript and your valuable comments that helped us to improve the final version. Below we provide a point-by-point response to your comments in sequential order.

Yours sincerely,

Lisa Nieuwboer, Julian Langowski

  1. This is a thoroughly interesting review that predicts that chytridiomycosis significantly affects tree frog adhesion and will thus be an important sublethal component of this disease in tree frogs.  My only disappointment is that the section on Bd-induced disruptions of tree frog attachment is theoretical and not based on actual data.  I have no reason to disagree with the conclusions of this section, but conclusions based on actual data would have significantly strengthened the overall conclusions.
  2. 2. All very interesting, but I wish you had some actual data!

We agree that the collection of data for assessment of the posed hypothesis is necessary. Therefore, we provide in the section “Perspectives” an outlook on how to collect such data. However, this is a review article and the implementation of novel data is thus out of the scope. We believe that this article provides a strong foundation for future studies on the hypothesized effects of chytridiomycosis on tree frog attachment. In fact, the authors are in the process of submitting a research proposal to further test the posed hypotheses, and to generate the data requested by the reviewer.

  1. Abstract last sentence:  I think the final conclusions are rather over-optimistic.  Certainly, the findings will increase our understanding of sub-lethal components of chytridiomycosis, but the design of mitigating strategies leading to the conservation of tree frog biodiversity seem to me, sadly, overoptimistic.

We thank the reviewer for pinpointing this overstatement, and modified this sentence to avoid overstating the direct impact on this work for the mitigation of chytridiomycosis.

  1. 2, 2nd paragraph: Isn’t the word ‘thallus’ restricted to plants?

The term thallus is not limited to plants and commonly used in the literature to describe the structure of Bd (PLoS one, Longcore et al., 1998, Mycologica, Piotrowski et al., 2003, Mycologica, Reeder et al., 2012).

  1. 1, 1st paragraph: You overemphasize the role of van der Waals forces (see Kappl et al., 2016, Bioinspiration & Bioimimetics) and omit the role of viscosity-dependent hydrodynamic forces (Federle et al., 2006).  The reference to mucus-mediated capillary adhesion is of course appropriate!

We agree that tree frog attachment likely relies on an intricate interplay of attachment mechanisms, including van der Waals forces as well as capillary forces. The current scientific literature reflects a lively debate on the individual contributions of these mechanisms to tree frog attachment. As this manuscript focusses on a potential link between tree frog attachment and chytridiomycosis, but not on the fundamentals of tree frog attachment per se, we refrain from further elaborations in this manuscript. Also, there is little evidence for the involvement of hydrodynamic forces, as shown by Emerson and Diehl (1980, Biological Journal of the Linnean Society). However, to provide a full overview of the mechanisms that have been proposed to be involved in tree frog attachment—including wet adhesive forces—we added a reference to a review by Endlein and Barnes (2015, Encyclopedia of nanotechnology) on this topic.

  1. 1, 4th paragraph: You discuss mechanisms that counteract detachment, yet such mechanisms need to be overcome each time the frog jumps or takes a step.

We agree that detachment is an important aspect of tree frog attachment and of bioadhesion in general, as explained in more detail in a recent review by Langowski et al. (2020, Integrative and Comparative Biology). Since our hypotheses on the effects of chytridiomycosis, however, largely focus on the phases of contact formation and contact maintenance, we refrain from further elaborations on toe pad detachment.

  1. 4, last paragraph: Add ref. 31 alongside refs 26 & 59 as it is a key paper.

Complied.

  1. Introduction, paragraph 5: Good point; a biomechanical study of the effects of chytridiomycosis on toe pad morphology and function should, indeed, produce useful conclusions on the sub-lethal effects of this disease in tree frogs.
  1. References:  The paper is well referenced.

Agreed.

Reviewer 3 Report

General Comments:

This very interesting and well-written review paper suggests a biomechanical mechanism by which sublethal chytridiomycosis could have population-level effects. In particular, the manuscript suggests several mechanisms by which adhesion of treefrog toe pads to substrate could be negatively affected by chytridiomycosis, and the potential behavioral (habitat use) and survival implications of loss of adhesion. The introduction sets up the review paper nicely, and the reviews of the biomechanics of treefrog toe pad adhesion and effects of chytridiomycosis on the amphibian epidermis logically lead to the primary tenet of the manuscript. The authors further go on to suggest the evidence and experiments necessary to test the hypotheses they generate, and summarize the far-reaching implications of such a line of research. The figures were very nicely done and supported the text well. Overall, this is a nice, thought-provoking review. My few specific comments below are minor grammatical suggestions.

Specific Comments:

Line 31:                      Insert “are” after “(16%).”

Line 210:                    Either delete “also” or move it to a position following “skin.”

Line 213:                    Insert “a” before “drainage.”

Line 269:                    Replace the last “of” with “on.”

Line 350:                    Delete “role.”

Author Response

Dear reviewer,

Thank you very much for your review of our manuscript “Does chytridiomycosis affect tree frog attachment?”. We also thank you for your kind words and pointing out several gramattical. Below we provide a point-by-point response to your comments.

Yours sincerely,

Lisa Nieuwboer, Julian Langowski

  1. Line 31: Insert “are” after “(16%).”
  2. Line 210: Either delete “also” or move it to a position following “skin.”
  3. Line 213: Insert “a” before “drainage.”
  4. Line 269: Replace the last “of” with “on.”
  5. Line 350: Delete “role.”

We implemented all suggested grammatical changes.